# Assessment of Processes to Increase the Useful Life and the Reuse of Reverse Osmosis Elements in Cape Verde and Macaronesia

**DOI:** 10.3390/membranes12060613

**Published:** 2022-06-13

**Authors:** Tomás Tavares, Jorge Tavares, Federico A. León-Zerpa, Baltasar Peñate-Suárez, Alejandro Ramos-Martín

**Affiliations:** 1Faculdade de Ciências e Tecnologia e da Escola de Ciências Agrárias e Ambientais, Universidade de Cabo Verde, Praia CP 279, Cape Verde; tomas.tavares@docente.unicv.edu.cv (T.T.); jorge.tavares@docente.unicv.edu.cv (J.T.); 2Departamento de Ingeniería de Procesos, Universidad de las Palmas de Gran Canaria, 35017 Las Palmas de Gran Canaria, Spain; alejandro.ramos@ulpgc.es; 3Departamento de Aguas, Instituto Tecnológico de Canarias, 35019 Santa Lucía, Spain; baltasarp@itccanarias.org

**Keywords:** reverse osmosis, membranes, desalination, reuse and recycling

## Abstract

Reverse osmosis membranes could be reused in the same or another desalination plant by replacing the membranes in the dirtiest first positions with those in the least damaged last positions, also changing the best first stage membranes to the second and vice versa. The useful life of these membranes could be extended by chemical cleaning and giving them a second life in tertiary treatment plants, as well as reusing them in industrial processes where special reverse osmosis membranes are used and degrade rapidly, in processes with leachates from landfill waste, and also an interesting option is the oxidation of reverse osmosis elements to obtain nanofiltration, ultrafiltration or microfiltration membranes for the elimination of physical dirt. The main categories of recycling by thermal processing commonly used in the industry include incineration and pyrolysis to produce energy, gas and fuel. These processes can be applied to mixed plastic waste, such as the combination of materials used in the manufacture of reverse osmosis membranes. Recycling of reverse osmosis elements from desalination plants is shown to be an opportunity, and pioneering initiatives are already underway in Europe. Energy recovery via incineration is feasible but is not considered in line with the environmental, social and political problems it may generate. However, the recycling of reverse osmosis elements via the pyrolytic industry for fuel production can be centralized in a new industry already planned in the Macaronesia area, and all obsolete osmosis membranes can be sent there. This is a technically and economically viable business opportunity with a promising future in today’s recycling market, as discussed in the article.

## 1. Introduction

This document develops a study on the evaluation of processes to increase the useful life and potential reuse of reverse osmosis membranes and components, especially in Cape Verde and in the whole Macaronesia cooperation area. Proven solutions for the reuse of membranes in tertiary treatment plants or as filters in industrial processes are addressed. Solutions for the reuse of effluent or by-products from other processes for the components are proposed. Results are obtained to increase the useful life of the elements, as well as the existence of by-products that open new opportunities for the waste or industrial sector. A qualitative and quantitative diagnosis of the membranes generated, especially in Cape Verde and Macaronesia in general, materials, periodicity, etc., is carried out.

This study is related to the circular economy and the blue economy, as well as the intersection of environmental aspects and material, economic and social resource use efficiency as drivers of growth and innovation for sustainable economic development [1,2,3].

Making seawater drinkable is one of the possible solutions to the shortage of drinking water. By desalinating seawater, fresh water suitable for water supply and irrigation can be obtained [4,5,6].

Desalination has placed Cape Verde on the world technological map of industrial water production, turning it into a large laboratory with the implementation of large-scale pilot plants, which developed a broad spectrum of existing desalination technologies over the years [3,4,5,6,7,8].

The growth in demand for water and the impossibility of forcing the use of natural resources further forces the development of water desalination, particularly in Cape Verde and in Macaronesia in general, to cover the water needs of the population, tourism, industry and agriculture [6,7,8,9].

At present, water desalination in Cape Verde and in Macaronesia in general allows us to reuse reverse osmosis elements from water desalination processes in such a way that they can help to improve and optimize the blue economy of these regions [10,11,12,13,14]. The DESAL+ living lab platform is creating knowledge related to desalination in Macaronesia and promoting in different initiatives the reuse of membranes (www.desalinationlab.com, accessed on 21 May 2022) [15].

## 2. Technical Actions to Be Carried Out

### 2.1. Re-Use of Reverse Osmosis Elements

Different actions for the reuse of membranes have been investigated in this study, and the following can be highlighted:In two-stage desalination plants, due to the fouling of the first-stage membranes compared to the second-stage membranes, it is proposed to replace the membranes before disposal and after cleaning. In this way, membranes could be reused and given a second life [15,16,17].Brackish water reverse osmosis membranes can also have a second life in tertiary treatment plants that do not require a very demanding quality of permeate for irrigation, especially if they are membranes with low fouling or with a large spacer, since the water coming from the treatment plant to the tertiary treatment plant is normally dirtier than the water from a brackish well installation, which is the most common way of collecting this type of water in Cape Verde in particular and in Macaronesia in general. Therefore, the membranes of the brackish water plant may have suffered salt precipitation, especially in the last positions of the second or third stage, but the best ones can be selected, and after a chemical cleaning they can be proposed for reuse in a tertiary treatment plant, where the problem, rather than salt precipitation, may be the fouling of the water. The membranes will eventually become clogged, but only after a second life and after they have been exploited 100%. Furthermore, the water from the reverse osmosis tertiary treatment plant will be used for irrigation or returned to the sea, such that it will not be used for drinking water because the law does not allow it; unlike in other countries such as Singapore, we do not run the risk that the reused membranes may have some kind of leak that could worsen the quality of the water and make it unsuitable for human consumption [15,18,19,20].There are some industrial processes that use special reverse osmosis membranes which end up very dirty in less than a month, so many of them must be replaced in a short time. By studying each industrial application that requires osmosis, depending on the salinity and pressures, it is possible to propose the reuse of second-hand membranes from other desalination plants in these processes to exploit them further before throwing them away. Other industrial processes include coffee concentrate, for example, where special reverse osmosis membranes are used and end up black in less than a month, meaning that many must be replaced in a short time. By studying each industrial application that requires osmosis, depending on salinity and pressures, it is possible to propose the reuse of second-hand membranes from other desalination plants in these processes to exploit them even more before throwing them away [15,21,22,23].There is also the possibility of leachate from landfill waste and other waste, which is sometimes treated by reverse osmosis to concentrate it; this process makes the membranes very dirty, and it is very difficult to find one that has a sufficiently long life to be profitable. In these cases, the option of reusing seawater membranes can be considered, if the salinity of the leachate is close to that of seawater, which is very common, or membranes from tertiary or brackish water plants if the salinity is lower. In this way, these reverse osmosis elements would be used in a final leachate concentration process before disposing of the membranes after they have been used 100%. Regarding leachate treatment, there are leachate evaporation systems in the complexes, although it has been accepted in WWTPs, which may be currently treating the surplus. However, it has always been thought, after consultation with experts in waste management, that the leachate could be used as a fertilizer after treatment. As salinity is one of the big problems, we think that treatment with membranes could be very interesting. There is bibliography on the subject, and it is believed that it could be an interesting line of work, even starting with a project [24,25,26,27].The membranes in a brackish water plant are more likely to be extended by chemical cleaning, but in these plants which usually have several stages, the membranes can also be combined between the two stages before disposal. Similarly, brackish water plants normally have a minimum of two or three stages to achieve conversions close to 75%, and this can help us to reuse older membranes in the third or sacrificial stage where they are more likely to precipitate salts earlier and be used to produce some more water until they are finally exploited to 100%.

A very practical and interesting option is the oxidation of reverse osmosis membranes, both seawater and chlorinated brackish water, to obtain nanofiltration membranes with lower salt rejection, ultrafiltration, or microfiltration for physical dirt removal in the line of cartridge filters [27].

Depending on the pore size, permeability and rejection capacity, polymeric membranes can be classified into microfiltration, ultrafiltration, nanofiltration and reverse osmosis. The rejection of these elements depends on the pore size, and by oxidizing them, we increase the pore diameter as shown in Table 1.

A pilot plant has been built to carry out validation about how the membrane is reused. Basically, it consists of the adaptation of a new pressure vessel for the validation of the reused membranes and to compare with original UF membranes, respectively. Each pressure vessel has a capacity for one membrane in series with the same mode of operation of the original plant, in terms of operating times, cleaning and flushing. The adaptation of the pressure vessel has basically consisted of a hydraulic modification of the frame, equipping pressure regulators at the entrance of the vessels and a system of measurement for the flow and pressure at the outlet of the permeate and rejection.

The purpose of this activity is to carry out the necessary tests to verify that the previously reused membranes can be used in various processes with different applications with each other and therefore validate the reusing of reverse osmosis membranes. It will allow the reinsertion in the market of what today is considered waste. Recycled membranes have been reused for: brackish water treatment, seawater treatment and wastewater treatment.

The case studied is about reused ultrafiltration membranes in the following way. Once removed, the polyamide layer of deteriorated reverse osmosis membranes, the layer exposed to filtration water, is composed of polysulfone, a more hydrophobic polymer than polyamide and therefore more susceptible to interact with the organic matter contained in the residual water. During the study carried out on the passive transformation of membranes of reverse osmosis on a laboratory scale, it was detected that it was possible to obtain membranes of ultrafiltration with a certain degree of residual polyamide on the surface of the membranes reused. Thus, to identify whether this fact favors the resistance of the membranes against fouling and/or improves the efficiency of the cleaning cycles, we carried out a study with transformed membranes (combining different times of exposure and concentration, and even different levels of exposure—30,000 and 300,000 ppm h). Some degree of residual polyamide was shown to improve the rejection capacity of the recycled ultrafiltration membranes and slightly favors the recovery of the permeability after applying cleaning cycles. The main advantage of this demonstrator is its use in a period of sufficiently long time, and in real conditions of variation of water quality, input and environmental conditions, temperature, etc. [27].

The initial objective of this process was to make a direct comparison on a pilot scale of membranes transformed (ultrafiltration) with commercial membranes of ultrafiltration in the brackish water treatment process, considering the possibility of later testing also with seawater and wastewater. For comparison of the efficiency of the ultrafiltration membranes, the same monitoring pilot plant was used to replicate the assays with transformed membranes using two types of commercial membranes (purchased specifically for the project). With monitoring, the plant’s initial comparison parameters were obtained, such as permeability, salt rejection, product water flow and ideal working pressure. It has been identified that two of the doses of exposure used in the reuse process allow for obtaining reused membranes with characteristics such as commercial ultrafiltration models. The objective of this action is to validate the membranes transformed in the process of water desalination pretreatment. This study has been developed because ultrafiltration has been imposing itself in recent years as an important alternative to conventional pretreatment (sand filtration) [27].

### 2.2. Recycling of Reverse Osmosis Components 

In this section, technical actions are described concerning the opportunities found in the recycling of reverse osmosis membrane components.

The treatment and recycling of solid plastic waste can be divided into four main categories: primary (re-extrusion), secondary (mechanical), tertiary (chemical) and quaternary (energy recovery). Primary recycling is generally carried out at the manufacturing plant by reintroducing clean waste into the extrusion cycle. This process cannot generally be applied to dirty waste products, such as clean reverse osmosis modules, as the recycling materials are not expected to reach the required quality. While only a small number of recycling options are directly applicable to the recycling of RO membranes, assessing their validity is an important step in the process of investigating all recycling opportunities.

During the mechanical recycling process, plastics are physically shredded to product size, separated from contaminants, washed and then used as raw material for the production of new products. Assimilable or incompatible polymers can cause deterioration of the mechanical property during the process, so for mechanical recycling to be economically viable, it is important that there is a large amount of clean and homogeneous single plastic polymer waste.

For membrane materials, each component must be considered individually to determine its potential suitability for mechanical recycling, assuming it can be successfully and economically separated. For example, the polypropylene feed spacer, has the capacity to be recycled directly using this method. In fact, polypropylene is commonly recycled in containers and packaging because of the strength and thermal and chemical resistances it can maintain, even after recycling. In Cape Verde, this can be done by the company CP5, S.A. for transformed plastics located on the island of Tenerife. Depending on the type of polyester components used, such as the permeate spacer, these also have the capacity to be mechanically recycled. Due to the nature of their polymer, ABS or noryl materials, such as the end caps and permeate tubes, may suffer from deterioration of physical properties when recycled by these methods, so they are generally reprocessed using other techniques. Finally, the flat membrane sheets, which make up a large proportion of the element, are constructed of different materials. In addition, membrane sheets can become contaminated with any kind of substance after prolonged use. Due to the nature of the process and the reasons mentioned above, direct mechanical recycling of the module can be prohibitively labor and cost intensive. This is because, there are many components together in the flat membrane (aromatic polyamide, polysulfone, polyester, glue, etc.) which, together with the physical fouling that can be added, as well as biofouling, salt fouling and so on, make it technically almost impossible to separate these materials at the source and cost excessive time and money.

Chemical (or feedstock) recycling is a process that breaks down plastic material into small molecules to be used as feedstock for petrochemical processes, using the method used to create the polymer chains, such as depolymerization and degradation. Polyester materials (as in permeate spacer and flat membrane components) are useful for chemical recycling processes, and hydrolysates are used to reverse the reaction of condensates used to make polymers, with the addition of water to the composition caused. Chemical recycling cannot typically be used with contaminated materials, and although it is cheaper and more complex than mechanical and primary recycling, its main advance is that heterogeneous polymers with limited use of pre-treatment can be processed. In this case, chemical recycling is more feasible than mechanical recycling for the treatment of reverse osmosis membranes and could be viable through a pyrolytic industry that is being planned for 2021 on the island of Tenerife [15].

These processes can be considered as viable recycling options. The main categories of thermal processing commonly used in industry include incineration; pyrolysis and thermal processing in the absence of oxygen; gasification, which is partial combustion with limited air to produce gas; and catalytic conversion to fuel. From an environmental point of view, gasification and hydrocarbons offer advantages over simple incineration as they produce fewer emissions, reduce waste and increase energy recovery. Most importantly, these processes can be applied to mixed plastic waste, such as the combination of materials used in the manufacture of reverse osmosis membranes. All these actions can make the membrane industry move towards a more circular economy [27].

## 3. Results

### 3.1. Results of the Qualitative and Quantitative Diagnostic Study of Reverse Osmosis Membranes

From this study, we obtained an approximate count of 1500 reverse osmosis membranes in operation in all the desalination plants in Cape Verde, with an annual replacement of around 300 membranes that would fall into disuse, an annual replacement rate of 20%. This estimate has been arrived at considering an average life of these reverse osmosis elements of 5 years, which is 2 years longer than the standard warranty of membrane manufacturers, which is 3 years. We have considered brackish water plants whose replacement is lower, and seawater plants with a higher replacement, especially large desalination plants that have an open intake collection and requirements of energy efficiency and water potability such as less than 1 mg/L of boron in the permeate, which currently require a higher replacement of membranes. Cape Verde also has the particularity that membranes are used more and last longer but still meet the above estimation. In general, membranes end up in landfills to be buried.

There is an estimated total of 687,000 m^3^/d of desalinated water production from reverse osmosis membrane desalination plants in Macaronesia, and approximately 660,000 m^3^/d in the Canary Islands, 20,000 m^3^/d in Cape Verde and 7380 m^3^/d in Porto Santo. Taking into account an average workflow depending on whether these facilities have open or well intakes, seawater, brackish water or tertiary treatment plants, the following quantity of membranes in disuse per year can be broken down by island, as shown in Table 2.

### 3.2. Results of the Reuse of Reverse Osmosis Membranes: Oxidation Process

The following results obtained from the oxidation of membranes for the reuse of reverse osmosis elements from desalination plants, as indicated in item 2.1, are highlighted.

Membrane oxidation has already been carried out and has worked successfully at the Barranco Seco WWTP of Emalsa (Las Palmas de Gran Canaria), so it has been considered for the reuse of reverse osmosis elements, and its technical and economic feasibility will be studied later to carry it out.The oxidation of reverse osmosis membranes must be carried out under the following dosage and exposure level (ppm × h) of free chlorine: 6200 (NF)—30,000 (UF). Before oxidation process in the pilot plant, it has been studied and already identified that the process of passive transformation at the laboratory scale is feasible, and the level of exposure of the membranes (ppm × h) to achieve a satisfactory transformation of the properties of deteriorated reverse osmosis membranes to properties of ultrafiltration has also been identified.

This action aims to transform damaged reverse osmosis membranes into ultrafiltration membranes. The flat sheet membranes of the reverse osmosis elements existing in the market are generally made up of three layers: a layer of aromatic polyamide that is in contact with water and acts as a selective barrier of 0.2 μm, supported by a sheet of microporous polysulfone which is usually an ultrafiltration membrane, and a structural polyester support. Polyamide is a polymer highly sensitive to exposure to oxidizing agents such as sodium hypochlorite. Taking advantage of this sensitivity, this chemical agent is used to degrade the layer of polyamide. If this degradation occurs partially, the membranes will have typical properties of ultrafiltration membranes. However, a complete degradation of polyamide gives rise to polysulfone membranes with properties typical of ultrafiltration membranes. The use of a brackish water solution (synthetic) allowed for determining the limits of transformation. It was observed that 30,000 ppm h (124 ppm × 242 h) were sufficient to obtain ultrafiltration membranes in terms of permeability and rejection. Membrane clipping flat sheets removed from commercial modules with spiral winding could be inserted into the membrane cell and tangential filtration conditions typical of the processes to be simulated. However, the times of exposure, especially to obtain recycled ultrafiltration membranes, turned out to be too long to be scalable at an industrial level, and in our case, to be used in transformation pilots. Thus, to optimize the times of transformation and make them more realistic, the combination of increasing the concentration of free chlorine and decreasing the times of exposure was studied, keeping the exposure level (ppm h) constant. It continued to fix an exposure level of 300,000 ppm h to transform the membranes into ultrafiltration [27].

The positive results of reusing reverse osmosis membranes by oxidizing them and transforming them into nanofiltration, ultrafiltration or microfiltration membranes are basically the following:
-The reuse of a reverse osmosis membrane has a lower environmental impact than the production of a new one. Water consumption during the membrane transformation process for membrane reuse is 20 times lower than the consumption to produce new membranes.-The carbon footprint of reused membranes in other processes is in the order of 40 to 60 times lower than the production of commercial membranes, and the price of reused membranes is 10 times lower than the price of new membranes. Therefore, in only 1 or 2 years of operation of reused membranes, the transformation process to reuse membranes is already beneficial both economically and environmentally.-The membrane oxidation process can be either actively in a pilot plant with pressure tubes or passively in a tank with several submerged membranes. From an economic and environmental point of view, passive processing of reverse osmosis membranes is better [15].

### 3.3. Results of Reverse Osmosis Membrane Recycling: Recycling and Pyrolysis Process

Similarly, this section includes the results of the recycling and recovery processes of reverse osmosis membranes through a pyrolytic process, also explained in item 2.2 above.

To collect the obsolete membranes in landfills in Cape Verde, it is necessary to pay, and this is cheaper than giving it another treatment outside Cape Verde, losing the opportunity to obtain a by-product from it such as energy recovery or fuels from processes such as pyrolysis.

The recycling of reverse osmosis elements does not exist in Cape Verde; it is a new business opportunity in the waste sector. In Europe, there is only one manager in Germany that recycles the membranes, but with the addition of transport, it is more expensive than taking them to landfill. Therefore, a considerable option would be to do the same recycling, but here in Cape Verde or the Macaronesia area.

On the other hand, we obtain other results through the pyrolysis process, which is a thermal degradation of a substance in the absence of oxygen, whereby these substances are decomposed by heat, without the combustion reactions taking place. The results of this process are as follows. The only oxygen present is that contained in the waste to be treated and the working temperatures are lower than those of gasification, ranging from 300 °C to 800 °C. As a result of the pyrolysis process, the following is obtained: gas, liquid residue, and solid residue with the following characteristics for each of them:The basic components of the gas are CO, CO_2_, H_2_, CH_4_ and more volatile compounds from the cracking of organic molecules, together with those already existing in the waste. This gas is very similar to the synthesis gas obtained in gasification, but there is a greater presence of tars, waxes, etc., to the detriment of gases, since pyrolysis works at lower temperatures than gasification. The gaseous products are a highly energetic fuel with a calorific value of approximately 50 MJ/m^3^.The liquid residue is basically composed of long-chain hydrocarbons such as tars, oils, phenols and waxes formed by condensation at room temperature.The solid waste consists of all non-combustible materials, which are either unprocessed or originate from molecular condensation with a high content of carbon, heavy metals and other inert components of these wastes.

Liquid and gaseous waste can be utilized by combustion in a steam cycle to produce electricity. Solid waste can be used as fuel in industrial plants, e.g., cement plants.

For energy recovery and incineration of the membranes, it is necessary to remove the encapsulated glass fiber from them, which is not difficult, and the rest of the elements that are very difficult to separate because they are glued and mixed are all plastic elements (aromatic polyamide, polysulfone, polyester, polyethylene, noryl, etc.) that can be incinerated to produce energy. Likewise, pyrolysis is a thermal treatment that mixes different plastics such as those that make up reverse osmosis membranes, as it has no PVC, metals or organic materials that would be harmful, thus obtaining high calorific value fuels as by-products of these elements.

There is no incinerator in Cape Verde, but it could be centralized on an island to collect all these reverse osmosis membranes and treat them. Likewise, there is no industry that carries out the pyrolysis process in Cape Verde, but one has already been planned in Tenerife (Plastics Energy) where all the reverse osmosis elements could be sent for this purpose. The technical and economic feasibility of these processes will also be studied later.

Regarding cogeneration at the Salto del Negro and Juan Grande Environmental Complexes (Gran Canaria), these consist of the use of the biogas generated in the biological processes and in the landfill, and therefore have nothing to do with the energy recovery of plastic waste [15].

The objectives of this action were the dimensioning of the membrane generator and receiver market recycled or reused, the analysis of the market size of the waste membrane generator and the evaluation of the receiving market for reused membranes. Regarding the evaluation of the recycling process and scaling potential, the following costs are compared:(a)The cost of the passive transformation would be between 28 and 38 EUR/module of ultrafiltration.(b)The cost for landfill disposal is in between 1 and 1.6 EUR/module.(c)This means a cost of 0.76–1.03 EUR/m^2^ for the transformed UF membranes.(d)The cost of the commercial UF membranes is around 17.2 EUR/m^2^.

Summarizing, the cost per m^2^ of reused membranes is between 18 and 19 times lower than the price of commercial membranes. After a technical, financial and economic analysis of the opportunities for the use of regenerated membranes (in comparison with new membranes, profitability, and opportunity costs), this would allow competitive prices to be applied (in between 50 and 80 EUR/module) with a short investment recovery period (less than 13 years) [27].

## 4. Conclusions

In addition to the great advantages of desalination, there are aspects that can be improved, such as the reuse of reverse osmosis elements in seawater, brackish and tertiary desalination processes.

Reuse of the reverse osmosis elements in the same plant or in tertiary plants with partial replacements, replacing the membranes in the dirtiest first positions with those in the least damaged last positions, is an economically viable alternative at a very low cost.

On the other hand, the oxidation of obsolete reverse osmosis membranes—to reuse, for example, the very damaged ones such as the first positions in the pressure tubes—gives us the opportunity to obtain a new by-product such as a cartridge filter (microfiltration), ultrafiltration or nanofiltration. In this sense, it shows the feasibility of betting on oxidation and selling the new membranes.

The recycling of reverse osmosis elements from desalination plants is shown as an opportunity and a pioneering initiative in Spain, with some experiences already contracted in Europe, where there is only one manager in Germany that recycles the membranes.

Energy recovery via incineration is feasible, but is not considered in view of the environmental, social and political problems it may generate.

On the other hand, for the recycling of reverse osmosis elements, the pyrolytic industry is a feasible way to send all obsolete osmosis membranes there.

This is a technically and economically viable business opportunity with a promising future in today’s recycling market.

## Figures and Tables

**Table 1 membranes-12-00613-t001:** Particle rejection as a function of pore size for different reverse osmosis, nanofiltration, ultrafiltration or microfiltration membranes.

Size	<0.001 µm	0.001–0.01 µm	0.01–0.1 µm	0.1–1 µm	1–10 µm
Separation Materials	Ion, Low Molecule Weight Organics	High Molecular Weight Polymer, Colloid	Colloid, Clay (Bacteria)	Clay (Coliform)	Clay (Cryptosporidium)
Water Treatments	Reverse Osmosis (RO)	Nanofiltration (NF)	Ultrafiltration (UF)	Microfiltration (MF)	Microfiltration (MF)
Types	RO/NF Membranes	RO/NF Membranes	Low Pressure Membranes	Low Pressure Membranes	Low Pressure Membranes

**Table 2 membranes-12-00613-t002:** Number of membranes, annual replacement, weight, and volume of membranes per island in Macaronesia.

Island	Number of Membranes in Use per Year	Annual Membrane Replacement	Weight (kg) Replacement	Volume (m^3^) Replacement
Lanzarote (Canary Islands)	8000	1600	32,000	67
Fuerteventura (Canary Islands)	8650	1730	34,600	73
Gran Canaria (Canary Islands)	23,500	4700	94,000	197
Tenerife (Canary Islands)	9350	1870	37,400	79
El Hierro (Canary Islands)	350	70	1400	3
La Gomera (Canary Islands)	150	30	600	1
Porto Santo (Madeira)	450	45	900	2
Praia-Palmarejo (Cape Verde)	700	100	2000	4
St Vincent (Cape Verde)	350	50	1000	2
Salt (Cape Verde)	350	50	1000	2

## Data Availability

The data presented in this study are available on request from the corresponding author.

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
