# Peer review of "Assessment of Processes to Increase the Useful Life and the Reuse of Reverse Osmosis Elements in Cape Verde and Macaronesia"

_membranes, 2022, doi:10.3390/membranes12060613_

Round 1

Reviewer 1 Report

I studied the paper and also compared it with earlier submission and review report. Authors made a minor changes in the manuscript. For instance, last paragraph of manuscript discusses the costs however, the presented data is not well organized. This part needs substantial revision. Table 3 lacks the information. I'll suggest authors to consult the previous review reports to improve the manuscript quality.

Author Response

Reviewer.

I studied the paper and also compared it with earlier submission and review report. Authors made a minor changes in the manuscript. For instance, last paragraph of manuscript discusses the costs however, the presented data is not well organized. This part needs substantial revision. Table 3 lacks the information. I'll suggest authors to consult the previous review reports to improve the manuscript quality.

Dear reviewer,

Firstly, thanks so much for your high support in this manuscript. Therefore, we strongly followed your comments and we got to improve the paper much more.

Regarding the last paragraph of the manuscript, you are right, therefore we have discussed the costs and presented data better organized than before to improve the understating of this issue. We haver reviewed it much more to improve it following good reviewer comments.

About table 3, you are right, it has been eliminated and the content has been explained accordingly in the body of the manuscript. Finally, we have also considered previous review reports to improve manuscript quality.

Thanks so much again for your help to improve our manuscript very much.

Reviewer 2 Report

The revised paper is very clear and easy to understand.

Author Response

Dear reviewer,

Thanks so much for your final approval and high support in this manuscript. We could improve it very much with your kind help.

Round 2

Reviewer 1 Report

Accept in current form

This manuscript is a resubmission of an earlier submission. The following is a list of the peer review reports and author responses from that submission.

Round 1

Reviewer 1 Report

This study is focused on the evaluation of different processes for recycling of RO membranes from circular economy view point. Subject is interested however there are so may scientific gaps in this study. Study is not supported by solid experimental back data. It is highly recommended to take in consideration the real experimental conditions. The used membranes carry multiple foulants that can significantly affect the recycling efficiency of different processes. However, this manuscript did not address this subject. Authors considered pyrolysis as a feasible process for membranes recycling compared with other methods. However, the higher energy demand and relevant direct and indirect environmental impacts does not show the cost-effectiveness of this process. Additionally, the weak part of this review is that it also does not cover the Life cycle assessment (LCA) and life cycle economics (LCE) methodologies for the membranes recycling.

Reviewer 2 Report

I would like to reject this work. First, who is the  corresponding author for this paper, I really don't know, at least from the manuscript. Also, this paper isn't well-prepared, from the abstract to the conclusion part. Finally, this kind of work should not be named as "article" since the research part of this paper is far too insufficient.

Reviewer 3 Report

In the relationship between chlorine concentration and exposure time in Table 3, the concentration in case A is specified in detail, but is there a serious concentration dependence? I think it would be easier to understand if the cases exposed to chlorine were unified only by the exposure level.

At the exposure times in Table 3, "." and "," are very confusing.

What will be the lifetime of the membrane when it is oxidized for use?

Is a reducing agent used when the membrane is oxidized with chlorine?

You state that reusing RO membranes will result in lower costs, but do you have quantitative, rather than qualitative, data?